# ESBL-Type and AmpC-Type Beta-Lactamases in Third Generation Cephalosporin-Resistant *Enterobacterales* Isolated from Animal Feces in Madagascar

**DOI:** 10.3390/ani14050741

**Published:** 2024-02-27

**Authors:** Ulrich Schotte, Julian Ehlers, Johanna Nieter, Raphaël Rakotozandrindrainy, Silver A. Wolf, Torsten Semmler, Hagen Frickmann, Sven Poppert, Christa Ewers

**Affiliations:** 1Department A—Veterinary Medicine, Central Institute of the Bundeswehr Medical Service Kiel, 24119 Kronshagen, Germany; ulrichschotte@bundeswehr.org (U.S.); johannanieter@bundeswehr.org (J.N.); 2Bernhard Nocht Institute for Tropical Medicine Hamburg, 20359 Hamburg, Germany; julian.ehlers@gmx.de; 3Department of Microbiology and Parasitology, University of Antananarivo, Antananarivo BP 566, Madagascar; rakrapha13@gmail.com; 4Genome Competence Centre, Robert Koch Institute, 13353 Berlin, Germany; wolfs@rki.de (S.A.W.); semmlert@rki.de (T.S.); 5Department of Microbiology and Hospital Hygiene, Bundeswehr Hospital Hamburg, 20359 Hamburg, Germany; frickmann@bnitm.de; 6Institute for Medical Microbiology, Virology and Hygiene, University Medicine Rostock, 18057 Rostock, Germany; 7Institute for Hygiene and Infectious Diseases of Animals, University of Giessen, 35392 Giessen, Germany

**Keywords:** cephalosporin resistance, fosfomycin, epidemiology, feces, animals, Madagascar

## Abstract

**Simple Summary:**

Little is known on the distribution of beta-lactam resistant Gram-negative bacteria in the gut of Madagascan animals. To increase respective knowledge, swabs from 49 animal stool droppings were collected in the Madagascan Tsimanapesotsa National Park and assessed by microbial culture, antimicrobial susceptibility testing, and sequence analysis. Third generation cephalosporine resistance, predominantly mediated by ampC-type beta-lactamases, was the most frequently observed beta-lactam resistance phenotype. Future studies should assess whether and to what extent an exchange of resistant isolates between humans and animals facilitates a local spread of resistant bacteria.

**Abstract:**

Third generation cephalosporin-resistant (3GCR) *Enterobacterales* are known to be prevalent in Madagascar, with high colonization or infection rates in particular in Madagascan patients. Extended spectrum beta-lactamases (ESBLs) have been reported to be the predominant underlying resistance mechanism in human isolates. So far, little is known on antimicrobial resistance and its molecular determinants in *Enterobacterales* and other bacteria causing enteric colonization of Madagascan wild animals. To address this topic, swabs from 49 animal stool droppings were collected in the Madagascan Tsimanapesotsa National Park and assessed by cultural growth of bacterial microorganisms on elective media. In addition to 7 *Acinetobacter* spp., a total of 31 *Enterobacterales* growing on elective agar for *Enterobacterales* could be isolated and subjected to whole genome sequencing. *Enterobacter* spp. was the most frequently isolated genus, and AmpC-type beta-lactamases were the quantitatively dominating molecular resistance mechanism. In contrast, the *bla*_CTX-M-15_ gene, which has repeatedly been associated with 3GC-resistance in Madagascan *Enterobacterales* from humans, was detected in a single *Escherichia coli* isolate only. The identification of the fosfomycin-resistance gene *fosA* in a high proportion of isolates is concerning, as fosfomycin is increasingly used to treat infections caused by multidrug-resistant bacteria. In conclusion, the proof-of-principle assessment indicated a high colonization rate of resistant bacteria in stool droppings of Madagascan wild animals with a particular focus on 3GCR *Enterobacterales*. Future studies should confirm these preliminary results in a more systematic way and assess the molecular relationship of animal and human isolates to identify potential routes of transmission.

## 1. Introduction

Similar to other African countries, third generation cephalosporin (3GC) resistance in *Enterobacterales* has relevantly increased in the course of the recent decades, even in severe infections like bacteremia [1]. In other types of infections, like urogenital tract infections [2,3,4] and wound infections [5], considerable resistance rates were also recorded with an increasing tendency during the last decade, next to varying, up to two-digit proportions of asymptomatic enteric carriage in Madagascans [6,7,8]. Higher rates of enteric colonization with 3GC-resistant (3GCR) *Enterobacterales* after hospitalization than before suggest ready transmission and/or selection under antibiotic pressure in the local healthcare setting [9]. Increased resistance rates in Madagascan *Enterobacterales* are particularly worrisome, because they have been described to play a predominant role in local neonatal infections [6,10,11] and thus affect a very vulnerable population. High rates of >70% for the acquisition of 3GCR *Enterobacterales* have been demonstrated for exposed Madagascan newborns [12].

Most Madagascan studies describe extended-spectrum beta-lactamases (ESBLs) as the major cause of 3GC-resistance in human *Enterobacterales* isolates [6,13,14,15], even in genera like *Enterobacter* spp. [6,13,15], in which inducible *ampC* beta-lactamases might have been expected to play a more dominant role. In clinical isolates, *bla*_CTX-M-15_ and *bla*_SHV-12_ have been described to be the regionally most abundant ESBL genes [15,16,17]. Only individual studies suggested a relevant proportion of inducible *ampC* mechanisms to account for relevant colonizing rates with 3GCR *Enterobacterales* on the skin and mucous membranes of Madagascan patients [18]. Reports on carbapenem resistance, mostly *bla*_NDM_-gene-associated, in Madagascan *Enterobacterales* have been published as well [3,8,17,19], although carbapenem resistance is less frequent there in comparison to 3GC-resistance. In comparison to *Enterobacterales*, acquired carbapenem resistance in Madagascan bacterial isolates has been described to be more relevant in non-fermentative Gram-negative rod-shaped bacteria like *Acinetobacter baumannii* [20].

Although the regional potential of animal sources of 3GCR *Enterobacterales* has been well recognized, studies on animal colonization in Madagascar are still scarce [21]. In regional livestock-associated ESBL-positive *Enterobacterales*, *bla*_CTX-M-1_ has been demonstrated as a quantitatively relevant resistance-associated gene [22].

To contribute to the so far scarce information on the carriage of 3GCR *Enterobacterales* and other antibiotic-resistant bacteria in Madagascan animals next to locally prevalent genetic resistance determinants, we screened for resistant bacteria in animal stool droppings in Madagascar and subjected resistant *Enterobacterales* and *Acinetobacter* spp. to whole genome sequencing. In silico analyses of genome sequences was performed to determine the presence of antimicrobial resistance (AMR) genes, virulence-associated genes, and multilocus sequence types. Genomic characteristics were further used to assess the pathogenic and zoonotic potential of bacterial strains obtained from animals in Madagascar.

## 2. Materials and Methods

### 2.1. Sample Collection and Ethical Permit

The sample collection took place in the coastal plain of south-west Madagascar close to the western border of the Tsimanampesotsa National Park. Within a three-months-interval from February to April 2017, animals were trapped with live traps (for details see Ehlers et al., 2019) [23] and brought to the research camp where fecal swabbings of a total of 49 stool droppings from *Echinops telfairi* (lesser hedgehog tenrec; *n* = 1), *Macronycteris commersoni* (Commerson’s roundleaf bat; *n* = 6), *Microcebus griseorufus* (reddish-gray mouse lemur; *n* = 32), *Pyxis arachnoides* (spider tortoise; *n* = 1), *Rattus rattus* (black rat; *n* = 6), *Setifer setosus* (greater hedgehog tenrec; *n* = 2), and *Triaenops menamena* (rufous trident bat; *n* = 1) were collected using GenoTube Livestock Swabs (Thermo Fisher Scientific, Waltham, MA, USA) enabling transport and storage at room temperature. The animals caught at the study site were subjected to sampling with no particular selection criteria; thus, the species distribution indirectly reflects their local abundance at the study site or at least their affinity to the applied live traps. Details on the locations of the sampling sites are shown in Figure 1. Information on the precise sampling dates and sampling coordinates is provided in Appendix A. The swabs were stored in a refrigerator at temperatures between 7 °C and 20 °C (depending on the power supply) until they were shipped to Germany for further analysis.

During the sampling procedure, compliance with all applicable institutional and/or national guidelines for the care and use of animals was assured. Ethical clearance for the field work and permit for the sample shipment were provided by the ethics committee of the Institute of Zoology of Hamburg University before the initiation of this study and authorized by the Ministère de l’Environnement, de l’Ecologie, de la Mer et des Forêts (Research permits: N°136/16/- and N°002/17/MEEF/SG/DGF/DSAP/SCB.Re; export permit: N°345-17/MEEF/SG/DGF/DREEFAAND/SFR).

### 2.2. Screening for Antibiotic Resistant Bacteria and Phenotypic Assessment

At the Department A—Veterinary Medicine, Central Institute of the Bundeswehr Medical Service Kiel, Kronshagen, Germany, the samples were subjected to cultural growth on agar plates selecting for *Enterobacterales* (CHROMagar Orientation, Mast Group Ltd., Bootle, UK) and *Acinetobacter* spp. (CHROMagar Acinetobacter, Mast Group Ltd.). To verify culturable bacterial background, all samples were additionally streaked on sheep blood agar and Gassner agar (Merck KG, Darmstadt, Germany). The screening for *Salmonella* spp. was done after selective enrichment in Rappaport–Vasiliadis medium (Merck KG, Germany), followed by streaking on XLD and BPLS agar (Sigma Aldrich, Darmstadt, Germany). Suspicious isolates were subjected to biochemical differentiation. For each different colony morphology that was visible with the bare eye, a single colony was chosen for further assessments. Additional species identification was done by mass spectrometry (MALDI Biotyper Version 3.1, DBUpdate V10.0.0.0, MBT Compass Library Revision L (2020), Bruker Daltonik, Bremen, Germany) as well as slide agglutination of presumptive *Salmonella* spp. (Thermo Fisher Scientific, Karlsruhe, Germany). The taxonomic status of *Enterobacter* spp. was additionally assessed at the whole genome level by performing average nucleotide identity (ANI) calculations based on BLAST+ using ANIb implemented in JSpeciesWS (© 2014–2023 Ribocon GmbH—Version: 4.0.2, Bremen, Germany).

Antimicrobial susceptibility testing (AST) was performed with the VITEK^®^2 system (BioMérieux, Nürtingen, Germany) using card AST-GN38 for *Enterobacterales* spp. and card AST-GN97 for *Acinetobacter* spp. according to the standards described in the CLSI document M100 [24]. As veterinary breakpoints are only available for some of the tested antibiotics and only for animals and indications not addressed in this study, minimal inhibitory concentration (MIC) data were interpreted according to CLSI document M100 (33rd edition) following breakpoints and intrinsic resistance data provided for *Enterobacterales* and *Acinetobacter* species of the *A. calcoaceticus–A. baumannii* (Acb) complex [24]. MICs for ceftiofur were interpreted according to CLSI document VET01S in case of *Enterobacterales* spp. [25] and as proposed in a recent publication in the case of *Acinetobacter* spp. [26].

### 2.3. Whole Genome Sequencing and Bioinformatic Analysis

Selected *Enterobacterales* with phenotypical resistance against 3GCR, as well as *Acinetobacter* spp. grown on CHROMagar *Acinetobacter* were sent for whole genome sequencing to the Institute for Hygiene and Infectious Diseases of Animals, Justus Liebig University of Giessen, Germany, in order to identify molecular AMR determinants and clonal lineages. Isolates were sequenced on an Illumina MiSeq platform using paired-end sequencing. After quality control using default parameters, adapter-trimmed reads were assembled using SPAdes v3.13.1 (Bankevich et al., 2012) [27]. Draft genomes were annotated using Prokka v1.13 (Seemann, 2014) [28]. Multilocus sequence types of *E. coli* (Achtman seven gene MLST scheme), *Enterobacter* spp., *Klebsiella* spp., *Salmonella* spp., and *A. baumannii* (Pasteur scheme) were identified in silico using MLST 2.0 provided by the Center for Genomic Epidemiology (https://cge.food.dtu.dk/services/MLST/, accessed on 6 January 2024).

### 2.4. Determination of Virulence-Associated Genes

Virulence genotyping of isolates belonging to the species *E. coli*, *Enterobacter* spp., *Salmonella enterica*, and *Klebsiella* spp. was performed by using VirulenceFinder 2.0 (https://cge.food.dtu.dk/services/VirulenceFinder/, accessed on 12 December 2023) hosted by the Center for Genomic Epidemiology and by using BacWGSTdb 2.0 (http://bacdb.cn/BacWGSTdb/Tools_results_single.php, accessed on 12 December 2023). The presence of the kleboxymycin biosynthetic locus in *Klebsiella* spp. was determined by using MyDbFinder 2.0 (https://cge.food.dtu.dk/services/MyDbFinder/, accessed on 6 January 2024) and the kleboxymycin nucleotide sequence of *K. oxytoca* strain MH43-1 (GenBank: MF401554.1).

## 3. Results

### 3.1. Species Differentiation and Antimicrobial Susceptibility Testing

The conducted growth on the chosen media led to the identification of 41 bacterial isolates (Table 1). In short, growth on agar selecting for *Enterobacterales* allowed the isolation of 21 *Enterobacter* spp. from *Rattus rattus* (2 isolates/6 animals, 33.3%), *Setifer setosus* (1/2, 50%), *Macronycteris commersoni* (2/2, 100%), *Triaenops menamena* (1/1, 100%), and *Microcebus griseorufus* (14/32, 43.8%). The ANI values of the *Enterobacter* spp. genomes against reference genomes and the core genome-based phylogeny revealed the presence of *E. hormaechei* ssp. *steigerwaltii* (*n* = 11), *E. hormaechei* ssp. *xiangfangensis* (*n* = 7), *E. hormaechei* ssp. *hormaechei* (*n* = 2), *E. cloacae* ssp. *cloacae* (*n* = 1), and *E. quasihormaechei* (*n* = 1). Five *Salmonella* spp. isolates were further obtained from *Setifer setosus* (2/2, 100%), *Echinops telfairi*, (1/1, 100%), and *Microcebus griseorufus* (2/32, 6.3%); two *Morganella morganii* were isolated from *Microcebus griseorufus* (2/32, 6.3%); and one *Escherichia coli*, one *Klebsiella oxytoca*, and one *Serratia marcescens* were isolated from *Microcebus griseorufus* (1 each/32, 3.1%). From selective media for *Acinetobacter* spp., five *A. radioresistens* isolates were obtained from *Microcebus griseorufus* (4/32, 9.4%) and from *Setifer setosus* (1/2, 50%), and one each of *A. baumannii* and *A. variabilis* were isolated from *Microcebus griseorufus* (3.1%).

Phenotypically obtained antimicrobial resistance data from the *Enterobacterales* spp. and *Acinetobacter* spp. are shown in Table 2 and Table 3.

### 3.2. Determination of Antimicrobial Resistance Genes

All 33 isolated *Enterobacterales* as well as seven *Acinetobacter* spp. were subjected to whole genome sequence analysis. The phenotypically observed third generation cephalosporin-resistance in the assessed *Enterobacter* spp. was most likely associated with class C ACT-type β-lactamases, which may naturally occur in these bacteria. Among 21 ACT type β-lactamases producing isolates, five known (3 × ACT-15, *n* = 3; ACT-16, *n* = 6; ACT-17, *n* = 3; ACT-56, *n* = 2; and ACT-115, *n* = 1) and five novel variants (ACT-120 to ACT-124) were assigned. The *E. quasihormaechei* isolate MG79-1 revealed a *bla*_ACT-59_-like gene variant which contained an internal stop codon, putatively encoding a non-functional enzyme. A novel variant of the AmpC β-lactamase family CMH was identified in *E. cloacae* ssp. *cloacae* strain MG59-6. Among 12 currently published CMH variants (CMH-1–CMH-12), the novel CMH-13 type showed highest amino acid nucleotide identity to CMH-3 (amino acid substitutions at positions 153 (Arg → Ser) and 218 (Asn → His).

Two novel DHA-type AmpC β-lactamases were observed in the two *Morganella morganii* isolates. Strain MG60-1 (IHIT45572) carried a new DHA variant (GenBank: WLO97155.1) with 99.21% amino acid identity to DHA-18 (NCBI Reference Sequence WP_063860101.1) that was originally identified in the species *M. morganii*. This new variant, termed DHA-31, differed from DHA-18 by three amino acids at positions 2 (Lys → Thr), 5 (Leu → Val), and 295 (Trp → Ser). The second novel DHA-variant (DHA-32) showed 99.47% amino acid identity to DHA-18 and showed the same amino acid changes at positions 2 and 5, while it was identical to the reference sequence at position 295. The only extended-spectrum β-lactamase gene observed in this study was *bla*_CTX-M-15_ that occurred in *E. coli* isolate MG59-3. The 3GC-suceptible *K. oxytoca* isolate carried a *bla*_OXY-2-18_-like gene, differing from the reference gene by a nucleotide change at position 213 (C to A), resulting in an amino acid change from aspartic acid to alanine. In accordance with the phenotypic AST results, isolates belonging to *S. enterica* and *Serratia marcescens* were negative for β-lactamase genes conferring extended cephalosporin resistance.

Intrinsic class-D beta lactamase genes *bla*_OXA-813_, *bla*_OXA-815_, and *bla*_OXA-816_ were identified among four *A. radioresistens* isolates, while one isolate carried a new oxacillinase gene of the OXA-23 family, which showed 98.53% amino acid identity to OXA-818 and was assigned as OXA-1221. A single *A. baumannii* isolate obtained in this study carried OXA-51-like gene *bla*_OXA-91_. Acquired *bla*_OXA_ genes capable of conferring resistance to carbapenems were not determined among the *Acinetobacter* spp. isolates from this study.

Few additional antimicrobial resistance determinants were also recorded as detailed in Table 4. Nearly all *Enterobacter* spp. isolates as well as the two *M. morganii* isolates, *S. marcescens* isolate MG91-1, and *K. oxytoca* isolate MG86-2 carried the fosfomycin-resistance gene *fosA.* An evolutionary analysis (Figure 2) revealed the assignment of *Enterobacter* spp. FosA proteins to the FosA2 family, which is correlated with a chromosomal location of the *fosA* gene. FosA proteins of *M. morganella* and *S. marcescens* isolates were genetically highly related or identical to FosA reference proteins of the same species, not clearly labelled with an allele number. The FosA protein of *K. oxytoca* isolate MG86-2 revealed the highest similarity to FosA5 (88.5%) and FosA10 (88.5%). Also for the non-*Enterobacter* sp. isolates, there was no indication for a plasmid location of *fosA*.

The chromosomally encoded aminoglycoside acetyltransferase *aac(6)-Iaa* resistance gene was present in all *Salmonella enterica* isolates. Only *E. coli* isolate MG59-3 showed an accumulation of AMR genes that are associated with resistance to broad-spectrum β-lactams (*bla*_TEM-1B_) aminoglycosides (*aph(6)-Id*, *aph(3″)-Ib*, and *mdf(A)*), sulfonamides (*sul2*), trimethoprim (*dfrA12*), quinolones (*qnrS1*), and tetracycline (*tet(A)*). All AMR genes except *bla*_TEM-1B_ and *mdf(A)* could be located on an IncFIB(K) plasmid that showed 99.2% identity to plasmid pCFSAN061766 (CP042872.1), which was identified in an *E. coli* strain from raw milk cheese in Egypt in 2016.

### 3.3. Determination of Multilocus Sequence Types

Multilocus sequence types (STs) were determined for those species where MLST schemes are publicly available. A total of 15 different STs were identified among 22 *Enterobacter* spp. isolates. Three *E. hormaechei* isolates isolated from a rat, a Commerson’s leaf-nosed bat, and a grey-brown mouse lemur revealed the already known ST987. Other STs that have previously been defined were observed in three *E. hormaechei* isolates from grey-brown mouse lemurs (ST1439, ST1461, and ST2412, respectively). Among 11 novel STs (ST2689–ST2699) determined in this study, most appeared as singletons, whereas ST2692 was identified in all *E. hormaechei* ssp. *xiangfangensis* isolates that were obtained from five grey-brown mouse lemurs and from a bat (*Triaenops menamena*).

The CTX-M-15-positive *E. coli* strain obtained in this study was assigned to ST7588 and phylogenetic group A. The single *K. oxytoca* isolate belonged to ST19 (clonal complex 2) which represents a highly prevalent and clinically relevant clonal group among human patients [29]. The five *Salmonella enterica* isolates were phylogenetically diverse as they were assigned to three known STs, including ST414, ST516, and ST3780, predicting serovars Give, Glostrup, and Eastborne, respectively, and two novel STs, namely ST10918 (predicted serological profile according to Enterobase: II Z:k:k) and ST10919 (predicted serological profile: II O:55:none:-).

Following the Pasteur scheme, the only *A. baumannii* isolate obtained in this study was assigned to ST2306, a so far rare sequence type that had initially been identified in a white stork (*Ciconia ciconia*) isolate from Poland in 2016 (GenBank: JAOWYX000000000.1).

### 3.4. Determination of Virulence-Associated Genes (VAGs)

Based on its VAG profile, the CTX-M-15-positive *E. coli* isolate could not be delineated to a distinct intestinal or extraintestinal pathogenic pathotype but rather resembled a non-pathogenic strain. In the *K. oxytoca* isolate MG86-2, WGS revealed the presence of various virulence genes, including genes for allantoin metabolism (*allA-allD*, *allR*, and *allS*), type 3 fimbriae (*mrkA*-*mrkC*), and siderophores, such as enterobactin (*entA*-*entC*, *entS fepA*, *fepD*, and *fepD*) and yersiniabactin (*ybtA*, *ybtE*, *ybtP*, *ybtQ*, *ybtS*, *ybtT*, *ybtU*, *ybtX*, *fuyA*, *irp1*, and *irp2*). Notably, the genome of isolate MG86-2 also contained the kleboxymycin biosynthetic gene cluster (BCG) that comprises 12 genes (*mfsX*, *uvrX*, *hmoX*, *adsX*, *icmX*, *dhbX*, *aroX*, *npsA*, *thdA*, *npsB*, *npsC*, and *marR*) and encodes for a tricyclic pyrrolobenzodiazepine that is associated with cytotoxicity in antibiotic-associated hemorrhagic colitis in humans [30]. The 17.430-bp locus of strain MG86-2 revealed 94.92% nucleotide sequence identity and 100% query coverage with the gene cluster of the *K. oxytoca* MH43-1 strain reference sequence (GenBank accession number MF401554.1). VAGs suggesting a virulence potential were also identified in the *A. baumannii* isolate. Among others, strain MG95-1 harbored genes encoding for the biosynthesis, efflux, and uptake of the major *Acinetobacter* siderophore acinetobactin (*basA*-*basD*, *basF*-*basH*, *basJ*, *bauB*-*bauF*), biofilm-associated locus genes *pgaABCD*, phospholipase C (*plc* and *plcD*), and Csu pilus genes (*csuABCDE*).

The five *Salmonella enterica* isolates harbored genes for various fimbrial operons, including *bcfABCDEFG*, *csgABCDEFG*, and *fimCDFHI*. Iron-acquisition-related aerobactin genes *iucABCD* and *iutA* were only present in the two unknown serotypes that were assigned to novel STs. In addition, several genes encoding for type three secretion system (T3SS) and T3SS effector proteins are more or less present in the *Salmonella* isolates. A detailed list of virulence gene profiles for all species included in VAG analysis is provided in Appendix A.

## 4. Discussion

This study was performed to contribute to epidemiological knowledge in the field of AMR in wild animals by assessing the enteric carriage of antibiotic-resistant bacteria in Madagascan animals. Our investigation revealed a number of novel data relevant to the field of AMR.

High proportions of intestinal colonization with antibiotic-resistant bacteria, in particular with 3GCR *Enterobacterales*, were found by cultural growth from the swabs of the stool droppings of different animal species. The observed high colonization rates with *Enterobacter* spp. matched previous findings from Madagascan human individuals [6,13,15,18]. However, while human isolates have been frequently associated with ESBL-type beta-lactamases in previous studies [6,13,15] with few exemptions [18], we could not confirm this finding for the *Enterobacter* spp. isolated from the animal stool samples. As expected, due to the abundance of inducible *ampC*-type beta-lactamases in *Enterobacter* spp. [31], the associated genes were considered as the most likely reasons for phenotypically observed 3GC-resistance in the obtained isolates from this study. The molecular resistance profile matches more a recent finding with *Enterobacter* spp.-colonized patients from the Madagascan Antananarivo hospital [18]. The *bla*_CTX-M-15_ gene as the underlying mechanism of 3GC-resistance in the single assessed *E. coli* isolate is typical for the Madagascan setting and well in line with previous regional reports [15,16,17]. The opportunistic pathogen *K. oxytoca* is naturally resistant to amino- and carboxypenicillins owing to low-level production of chromosomal β-lactamases of the OXY group [32]. Overproduction of *bla*_OXY_-genes in *K. oxytoca* due to promoter-up mutations results in reduced susceptibility or even resistance to other β-lactams, such as penicillin-inhibitor combinations, cefuroxime and cefotaxime and appears in approximately 10–20% of clinical isolates [32]. Our wild animal isolate harbored an OXY-2-8-like β-lactamase without a mutation in the −10 hexamer chromosomal region (−10 region: (A)GATAGT), which is in line with its 3GC-susceptible phenotype. Intrinsic class-D beta lactamase genes *bla*_OXA-813_, *bla*_OXA-815_, and *bla*_OXA-816_ were identified among four *A. radioresistens* isolates, while one isolate carried a new oxacillinase gene of the OXA-23 family, which was defined as OXA-1221. A single *A. baumannii* isolate obtained in this study carried the OXA-51-like gene *bla*_OXA-91_. Acquired *bla*_OXA_ genes were not determined.

Overall, the low abundance of further molecular antimicrobial resistance (AMR) determinants, as shown in Table 4, suggests the isolation of environmental isolates with low antimicrobial selection pressure due to exposure to antimicrobial drugs. However, the high percentage of isolates carrying a *fosA* gene, which confers resistance to fosfomycin, was not to be expected. This old antibiotic regained relevance in clinical practice for the treatment of complicated infections caused by multidrug-resistant bacteria [33]. Therefore, the emergence of *fosA* among rather naïve wild animal populations warrants further investigations, as it might have a significant impact on public health.

The CTX-M-15-producing *E. coli* strain MG59-3 from a grey-brown mouse lemur revealed a number of additional AMR genes that might confer resistance to aminoglycosides, sulfonamides, and tetracyclines. Although the strain also harbored the plasmid-mediated quinolone-resistance (PMQR) gene *qnrS1*, we could not determine phenotypic resistance to fluoroquinolones. FQ-resistance is mostly due to chromosomal mutations that alter the drug target enzymes DNA gyrase (topoisomerase II) and (*gyrA* and *gyrB*) and DNA topoisomerase IV (*parC* and *parE*) [34]. Our strain lacked mutations in these regions, but the presence of *qnrS1* most likely conferred the observed increased MIC to enrofloxacin (1 mg/L). The multidrug-resistant phenotype of strain MG59-3 indicates that wild animals might acquire AMR bacteria and/or genes from sources that are under anthropogenic influence. The Enterobase strain database (https://enterobase.warwick.ac.uk/species/ecoli/search_strains; accessed on 21 December 2023) contains 21 *E. coli* strains from various sources (e.g., environment, livestock, and wild animals) and countries (e.g., Kenya, Pakistan, United Arab Emirates, and USA) that belong to the same sequence type as our strain, namely ST7588. Interestingly, one strain from an unspecified source originated from the Antananarivo area in Madagascar and showed an AMR gene profile that was highly similar to that of strain MG59-3. Like our strain, *E. coli* C32b187a harbored *bla*_CTX-M-15_, *bla*_TEM-1B_, *aph(6)-Id*, *dfrA14*, *mdf(A)*, and *tet(A)*, while it lacked *aph(3″)-Ib*, *qnrS1*, and *sul2*. Moreover, a core-genome-based MLST comparison performed with BacWGSTdb (http://bacdb.cn/BacWGSTdb/Tools.php; accessed on 21 December 2023) further revealed a high similarity (30 different alleles) of strain MG59-3 with two ST7588 *E. coli* strains isolated from cattle in the USA in 2016. One of these strains (KCJK6915; Acc.-No. SRNT01) even showed an identical AMR gene profile. Thus, although we observed MG59-3 as a single isolate, the phylogenetic grouping and AMR gene profile indicate a wider distribution of this clonal type, which might be due to transmission events taking place between environmental sources, livestock and wild animals.

We also detected one *K. oxytoca* isolate (MG86-2) from a bat. After *K. pneumoniae*, *K. oxytoca* is the second most common *Klebsiella* species causing diseases in humans, such as pneumonia, urinary tract infection and skin infections [35]. As an opportunistic pathogen, it can also act in the dysbiotic human intestinal microbiota causing antibiotic-associated hemorrhagic colitis (AAHC) [30]. The single *K. oxytoca* isolate obtained in this study could be assigned to ST19 that has been reported as highly prevalent among clinical isolates from the UK [29] and has recently been associated with AAHC in a human patient in Austria [36]. Of note, it harbored the biosynthetic gene cluster encoding for a tricyclic pyrrolobenzodiazepine, which has been associated with cytotoxicity in AAHC caused by *K. oxyotoca* [30], suggesting a novel source of this molecular determinant in wild animals in Madagascar.

The only *A. baumannii* isolate (MG95-1) obtained in this study was cultured from the feces of a grey-brown mouse lemur. It was assigned to ST2306, a sequence type that was originally detected in an isolate from a white stork (*Ciconia ciconia*) in Poland in 2016 and that was reported in a recent study in a multidrug-resistant *A. baumannii* clinical strain [37]. The detected actinetobactin virulence cluster is considered one of the most relevant mechanisms mediating virulence of *Acinetobacter baumannii* isolates [38]. Further, the isolate encoded factors critical for biofilm formation [39], including genes for CSU pili [40] as well as the cytolytic factor phospholipase C [41], stressing its potential etiological relevance in case of nosocomial transmission to susceptible hosts as well.

The lacking assignability of the majority (72.7%) of *Enterobacter* spp. isolates to known multilocus sequence types confirms that they are most likely different from typical hospital-associated clones or those associated with clinical diseases in humans. Except for β-lactamase genes, other AMR genes were only rarely detected, including the fosfomycin gene *fosA* that occurred in 95.5% of the isolates, suggesting the presence of a rather antimicrobial naïve population of *Enterobacter* spp. isolates among the wild animals studied. In contrast, at least three of the five *Salmonella enterica* isolates could be assigned to sequence types that have previously been observed in isolates from humans, livestock, and wild animals (ST516; SS01-3 from greater hedgehog tenrec), from wild animals and food (ST3780, SS02-5 from greater hedgehog tenrec), and from humans and animals (ST414, ET01-4 from lesser hedgehog tenrec). *Salmonella* Give ST516 was one of the dominant sequence types identified in a study from Lagos, Nigeria (12 from humans, 2 from poultry and cattle, 2 from sewage samples), underlining the worldwide distribution of this non-typhoidal *Salmonella* serovar [42]. While fluoroquinolone resistance was common in *Salmonella* Give in the previous African study [41], such resistance was not observed in our assessment. Of note, the detection of aminoglycoside resistance genes in the salmonellae isolates in our study, however, confirm the high phenotypic aminoglycoside resistance rate previously reported [42].

The novel ST10918 is a double locus variant of ST7327. The Enterobase database contains two human clinical isolates of ST7327 from the UK and Canada. Interestingly, the second novel ST detected in this study from a grey-brown mouse lemur is phylogenetically related to ST1122 (double locus variant). Only one ST1122 isolate is listed in the Enterobase database, and this was isolated from a wild animal, namely a reptile, from Madagascar in the year 1962. Although none of our wild animal isolates carried genes typically located on *Salmonella* virulence plasmids (*spv* genes), most isolates possessed VAGs specific for *Salmonella* pathogenicity islands (Appendix A). *Salmonella* infections still represent an important public health issue worldwide and non-typhoidal *Salmonella* (NTS) have previously been associated with bloodstream infections and gastroenteritis, especially in children in Sub-Saharan Africa [42]. Next to genes for the type III secretion system, which is considered as a main element of salmonellae-associated virulence [43], fimbrial genes associated with persistence of salmonellae in the mammal intestine could be recorded [44].

Altogether, the proportion of detected AMR bacteria within the assessed animal stool samples is low to moderate and can thus be well explained by occasional contacts with the Madagascan human civilization that is much more severely affected by the resistance issue. In particular, only individual cases of ESBL-type resistance were detected in the animal stool samples of this study, while the ESBL mechanism has been described to account for high colonization rates with third generation cephalosporin-resistant *Enterobacterales* in both Madagascan people and livestock [6,13,14,15,16,17,21,22]. The considerably higher rates of colonization with *ampC*-positive *Enterobacter* spp. are well in line with observations in Madagascan patients and healthcare workers, in which this genus-resistance type-combination accounted for a major part of recorded 3GCR in isolated colonizing *Enterobacterales* [17]. In this regard, their regional abundance seems to be typical for Madagascar. Finally and with focus on the isolated *Acinetobacter* spp., recorded resistance profiles were close to the wild-type situation, and in particular, carbapenem resistance-mediating genes like those previously described for the Madagascan setting were not recorded. Altogether, the observed resistance profiles match the expectations from the literature quite well.

This study has a few limitations. The major limitations are the relatively low sample size, the more or less arbitrary sampling pattern, and the non-selective culturing approach. As such, the assessment provides only preliminary information and cannot replace future systematic analyses. The selective culturing of samples on media containing cefotaxime and ceftazidime might have increased the number of isolated AMR bacteria. Thus, the data cannot provide a real estimate for the distribution of ESBL- and AmpC-β-lactamase-producing bacteria among samples from the given animal population. Considering the still scarcely available epidemiological information on environmental isolates from the assessed region and with respect to the detailed molecular characterization provided for the obtained *Enterobacterales* and *Acinetobacter* isolates, the assessment may nevertheless serve as a useful proof-of-principle. However, it is still interesting to isolate bacterial species of clinical relevance from wild mammals, for which epidemiological reports from Madagascar are widely missing.

## 5. Conclusions

The presented proof-of-principle assessment indicated high enteric colonization rates of Madagascan mammals with resistant bacteria. In particular, 3GCR *Enterobacterales* with *ampC*-type beta-lactamases as the underlying resistance mechanism were found to be highly prevalent. In addition, the identification of the fosfomycin-resistance gene *fosA* in a significant number of isolates is a cause for serious concern and warrants further investigation. Due to the expected low antibiotic pressure, natural colonization is assumed, showing the general occurrence of antibiotic resistance genes in nature. Future systematic screening should be conducted to confirm the findings and further, molecular comparisons of colonizing resistant bacteria in animals and human individuals seem advisable to assess potential transmission routes.

## Figures and Tables

**Figure 1 animals-14-00741-f001:**
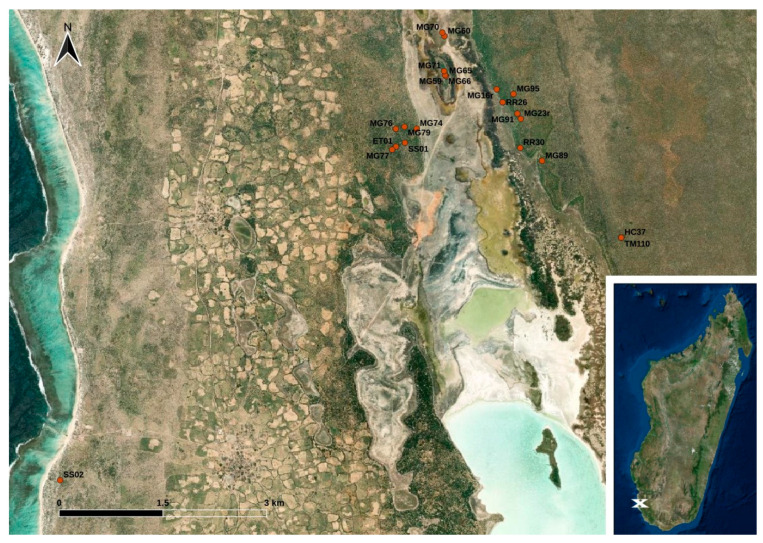
Map of the study site located in south-west Madagascar. The dots indicate the trapping points of the animals (see Appendix A) (designed with QGis based on Esri satellite imagery data (https://services.arcgisonline.com/ArcGIS/rest/services/World_Imagery/MapServer, accessed on 12 December 2023)).

**Figure 2 animals-14-00741-f002:**
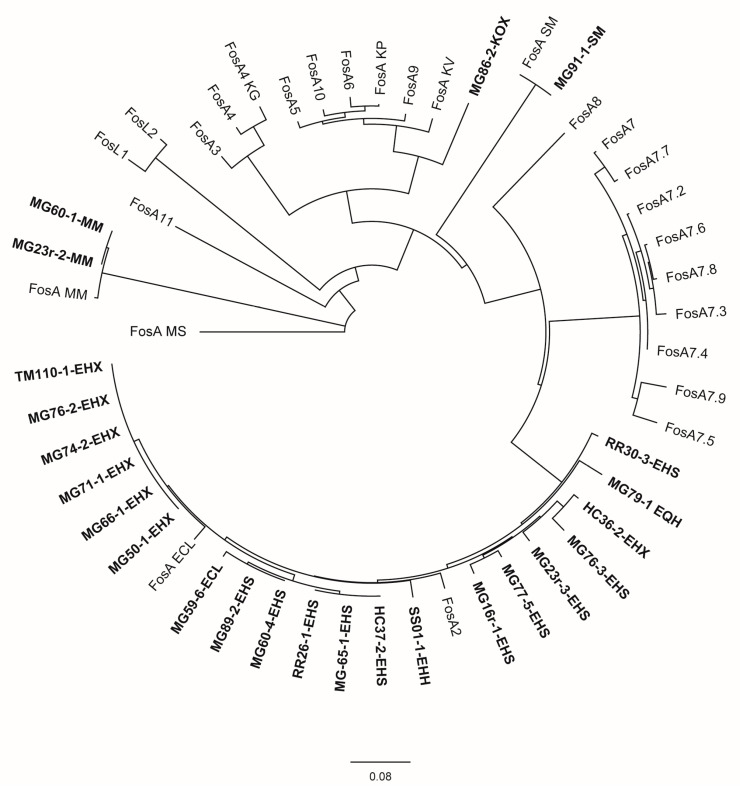
Evolutionary analysis and phylogenetic tree of FosA/C2/L1-2 proteins available in GenBank for *Enterobacterales* and FosA proteins identified in this study inferred by applying the maximum likelihood method using Geneious Prime^®^ 2023.2.1 GenBank Accession numbers of public sequences are given in parentheses below. Study isolates are written in bold. Isolate MG59-4-EHH was excluded from the analysis due to an interruption of the *fosA* gene by an IS911 element. ECL = *E. cloacae* ssp. *cloacae*, EHH = *E. hormaechei* ssp. *hormaechei*, EHX = *E. hormaechei* ssp. *xiangfangensis*, EHS = *E. hormaechei* ssp. *steigerwaltii*, EQH = *E. quasihormaechei*, KG = *Kluyvera georgiana*, KOX = *K. oxytoca*, KV = *K. variicola*, MM = *M. morganii*, MS = multispecies, SM = *S. marcescens*; FosA ECL (AWG41971.1), FosA KP (CDO16183.1), FosA KV (AWG41960.1), FosA MM (VDY35025.1), FosA MS (WP_154635598.1), FosA SM (QOW96986.1), FosA2 (WP_025205684.1), FosA3 (WP_014839980.1), FosA4 (WP_034169466.1), FosA4 KG (WP_064548962.1), FosA5 (WP_012579083.1), FosA6 (WP_069174570.1), FosA7 (WP_000941934.1), FosA7.2 (WP_000941935.1), FosA7.3 (WP_023231494.1), FosA7.4 (WP_023216493.1), FosA7.5 (WP_000941933.1), FosA7.6 (WP_061377147.1), FosA7.7 (WP_058653118.1), FosA7.8 (WP_079820715.1), FosA7.9 (WP_071684814.1), FosA8 (WP_063277905.1), FosA9 (WP_114473955.1), FosA10 (WP_004214174.1), FosA11 (QZL11398.1), FosL1 (WP_161667239.1), FosL2 (WP_188331883.1).

**Table 1 animals-14-00741-t001:** Association between obtained bacterial isolates and animal fecal samples.

Bacterial Isolates (*n*)	Origin (*n* Isolates/*n* Animals, %)
*Enterobacter* spp. (*n* = 21)	*Rattus rattus* (2/6, 33.3%), *Setifer setosus* (1/2, 50%), *Macronycteris commersoni* (2/2, 100%), *Triaenops menamena* (1/1, 100%), *Microcebus griseorufus* (14/32, 43.8%)
*Salmonella* spp. (*n* = 5)	*Setifer setosus* (2/2, 100%), *Echinops telfairi*, (1/1, 100%), *Microcebus griseorufus* (2/32, 6.3%)
*Morganella morganii* (*n* = 2)	*Microcebus griseorufus* (2/32, 6.3%)
*Escherichia coli* (*n* = 1)	*Microcebus griseorufus* (1/32, 3.1%)
*Klebsiella oxytoca* (*n* = 1)	*Microcebus griseorufus* (1/32, 3.1%)
*Serratia marcescens* (*n* = 1)	*Microcebus griseorufus* (1/32, 3.1%)
*A. radioresistens* (*n* = 5)	*Microcebus griseorufus* (4/32, 12.5%) *, *Setifer setosus* (1/2, 50%)
*Acinetobacter baumannii* (*n* = 1)	*Microcebus griseorufus* (1/32, 3.1%)
*Acinetobacter variabilis* (*n* = 1)	*Microcebus griseorufus* (1/32, 3.1%)

* Comprised two different colony morphologies from a single animal.

**Table 2 animals-14-00741-t002:** Results of antimicrobial susceptibility testing of *Enterobacterales* isolates (MIC data are provided as mg/L).

Strain Name *	Species (WGS)	AMP	AMC	PIP	CEF	CPD	CTF	IPM	TET	NIT	CHL	ENR	MAR
*Enterobacter* species													
RR26-1	*Ent. horm.* ssp. *steigerwaltii*	≥32	16	8	≥64	≥8	≥8	≤1	≤1	≥512	≥64	≤0.12	≤0.5
RR30-2	*Ent. horm.* ssp. *steigerwaltii*	≥32	≥32	8	≥64	≥8	≥8	≤1	2	64	8	≤0.12	≤0.5
HC37-2	*Ent. horm.* ssp. *steigerwaltii*	≥32	≥32	≤4	≥64	2	2	≤1	≤1	64	4	≤0.12	≤0.5
MG16r-1	*Ent. horm.* ssp. *steigerwaltii*	≥32	≥32	≤4	≥64	2	2	2	≤1	≤16	≤2	≤0.12	≤0.5
MG23r-3	*Ent. horm.* ssp. *steigerwaltii*	≥32	≥32	≥128	≥64	≥8	≥8	≤1	≤1	≤16	4	≤0.12	≤0.5
MG60-4	*Ent. horm.* ssp. *steigerwaltii*	≥32	≥32	≤4	≥64	≥8	4	≤1	≤1	64	4	≤0.12	≤0.5
MG65-1	*Ent. horm.* ssp. *steigerwaltii*	≥32	≥32	≥128	≥64	≥8	≥8	≤1	2	32	4	≤0.12	≤0.5
MG76-3	*Ent. horm.* ssp. *steigerwaltii*	≥32	≥32	≤4	≥64	≥8	4	≤1	≤1	32	8	≤0.12	≤0.5
MG77-5	*Ent. horm.* ssp. *steigerwaltii*	≥32	≥32	8	≥64	≥8	4	≤1	2	64	8	≤0.12	≤0.5
MG89-2	*Ent. horm.* ssp. *steigerwaltii*	≥32	≥32	≥128	≥64	≥8	≥8	≤1	2	32	4	≤0.12	≤0.5
MG91-5	*Ent. horm.* ssp. *steigerwaltii*	≥32	≥32	8	≥64	≥8	≥8	≤1	2	64	16	≤0.12	≤0.5
HC36-2	*Ent. horm.* ssp. *xiangfangensis*	16	≥32	≤4	≥64	1	≤1	≤1	≤1	64	4	≤0.12	≤0.5
TM110-1	*Ent. horm.* ssp. *xiangfangensis*	≥32	≥32	8	≥64	≥8	≥8	≤1	≤1	128	16	≤0.12	≤0.5
MG66-1	*Ent. horm.* ssp. *xiangfangensis*	≥32	≥32	≥128	≥64	≥8	≥8	≤1	≤1	128	16	≤0.12	≤0.5
MG70-1	*Ent. horm.* ssp. *xiangfangensis*	≥32	≥32	8	≥64	≥8	≥8	≤1	≤1	64	16	≤0.12	≤0.5
MG71-1	*Ent. horm.* ssp. *xiangfangensis*	≥32	≥32	8	≥64	≥8	≥8	≤1	≤1	128	16	≤0.12	≤0.5
MG74-2	*Ent. horm.* ssp. *xiangfangensis*	≥32	≥32	≤4	≥64	2	2	≤1	≤1	64	8	≤0.12	≤0.5
MG76-2	*Ent. horm.* ssp. *xiangfangensis*	≥32	≥32	16	≥64	≥8	≥8	≤1	≤1	128	8	≤0.12	≤0.5
SS01-1	*Ent. horm.* ssp. *hormaechei*	≥32	≥32	≤4	≥64	1	≤1	≤1	≤1	64	4	≤0.12	≤0.5
MG59-4	*Ent. horm.* ssp. *hormaechei*	≥32	≥32	≤4	≥64	≥8	4	≤1	≤1	32	4	≤0.12	≤0.5
MG59-6	*Ent. cloacae* ssp. *cloacae*	≥32	≥32	≥128	≥64	≥8	≥8	≤1	≤1	64	8	≤0.12	≤0.5
MG79-1	*Ent. quasihormaechei*	≥32	≥32	8	≥64	≥8	≥8	≤1	2	64	16	≤0.12	≤0.5
Other *Enterobacterales* species													
MG59-3	*Escherichia coli* *	≥32	8	≥128	≥64	≥8	≥8	≥1	≥16	≤16	8	1	≤0.5
MG86-2	*K. oxytoca*	≥32	8	16	≤4	≤0.25	≤1	≤1	≤1	32	≤2	≤0.12	≤0.5
MG23r-2	*Morganella morganii*	≥32	≥32	≤4	≥64	2	≤1	8	≤1	128	4	≤0.12	≤0.5
MG60-1	*Morganella morganii*	≥32	≥32	≤4	≥64	0,5	≤1	8	≤1	64	≤2	≤0.12	≤0.5
SS01-3	*Salmonella enterica*	≤2	≤2	≤4	8	≤0.25	≤1	≤1	≤1	32	8	≤0.12	≤0.5
SS02-5	*Salmonella enterica*	≤2	≤2	≤4	8	≤0.25	≤1	≤1	≤1	32	4	≤0.12	≤0.5
ET01-4	*Salmonella enterica*	≤2	≤2	≤4	8	≤0.25	≤1	≤1	≤1	≤16	8	≤0.12	≤0.5
MG23r-5	*Salmonella enterica*	≤2	≤2	≤4	8	≤0.25	≤1	≤1	≤1	32	8	≤0.12	≤0.5
MG77-6	*Salmonella enterica*	≤2	≤2	≤4	8	≤0.25	2	≤1	≤1	64	8	≤0.12	≤0.5
MG59-1	*Serratia marcescens*	16	≥32	≤4	≥64	≤0.25	≤1	≤1	≥16	256	8	0.25	≤0.5
MG91-1	*Serratia marcescens*	16	≥32	≤4	≥64	0.5	≤1	≤1	≥16	256	8	0.5	≤0.5

* Strain name includes abbreviation of animal source, sample I.D., and isolate number. ET = *Echinops telfairi*; HC = *Hipposideras commersoni*; MG = *Microcebus griseorufus*; RR = *Rattus rattus*; SS = *Setifer setosus*; TM = *Triaenops menamena.* AMP = ampicillin; AMC = amoxicillin/clavulanate; CEF = cefalexin (1st gen. cephalosporin); CHL = chloramphenicol; CPD = cefpodoxime (3rd gen. cephalosporin); CTF = ceftiofur (3rd gen. cephalosporin); ENR = enrofloxacin; IPM = imipenem; MAR = marbofloxacin; NIT = nitrofurantoin; PIP = piperacillin; TET = tetracycline. Yellow-shaded MIC values: resistant according to expert rules for intrinsic resistances provided by CLSI; red- and light red-shaded MIC values: resistant and intermediate phenotype according to CLSI breakpoints; MICs for ceftiofur were interpreted as proposed previously [26] according to breakpoints for 3rd generation cephalosporines set by CLSI for *Acinetobacter* spp.

**Table 3 animals-14-00741-t003:** Phenotypic resistance results (MIC value) of the *Acinetobacter* spp.

Strain Name *	Species	Animal Source	AMP	AMC	CEF	CFT	CTF	AMK	GEN	IPM	ENR	MAR	TET	DOX	CHL	SXT
MG95-1	*A. baumannii*	MG	16	4	≥64	≥64	≥8	≤2	≤1	≤0.25	≤0.12	≤0.5	≤1	≤0.5	≥64	≤20
SS02-1	*A. radioresistens*	SS	4	≤2	≤4	32	4	≤2	≤1	≤0.25	≤0.12	≤0.5	≤1	≤0.5	8	≤20
MG60-2	*A. radioresistens*	MG	≤2	≤2	≤4	16	≥8	≤2	≤1	≤0.25	≤0.12	≤0.5	≤1	≤0.5	8	≤20
MG60-3	*A. radioresistens*	MG	8	4	16	32	≥8	≤2	≤1	≤0.25	≤0.12	≤0.5	≤1	≤0.5	16	≤20
MG87-2	*A. radioresistens*	MG	16	≤2	8	≥64	≥8	≤2	≤1	≤0.25	≤0.12	≤0.5	≤1	≤0.5	8	≤20
MG91-2	*A. radioresistens*	MG	16	≤2	≤4	16	4	≤2	≤1	≤0.25	≤0.12	≤0.5	≤1	≤0.5	16	≤20
MG77-2	*A*. *variabilis*	MG	≤2	≤2	≤4	16	4	≤2	≤1	≤0.25	≤0.12	≤0.5	≤1	≤0.5	≤2	≤20

* Strain name includes abbreviation of animal source, sample I.D., and isolate number. MG = *Microcebus griseorufus*; SS = *Setifer setosus*; AMP = ampicillin; AMC = amoxicillin/clavulanate; AMK = amikacin; CEF = cefalexin (1st gen. cephalosporin); CFT = cefotaxime (3rd gen. cephalosporin); CHL = chloramphenicol; CTF = ceftiofur (3rd gen. cephalosporin); DOX = doxycycline; ENR = enrofloxacin; GEN = gentamicin; IPM = imipenem; MAR = marbofloxacin; SXT = trimethoprim/sulfamethoxazole; TET = tetracycline. Yellow-shaded MIC values: resistant according to expert rules for intrinsic resistances provided by CLSI; red-shaded MIC values: resistant phenotype according to CLSI breakpoints; MICs for ceftiofur were interpreted as proposed previously [26] according to breakpoints for 3rd generation cephalosporines set by CLSI for *Acinetobacter* spp.

**Table 4 animals-14-00741-t004:** Multilocus sequence types and genotypic resistance of *Enterobacterales* and other gram-negative species isolated in this study.

Strain Name *	Species	Multilocus Sequence Type **	AMR Genes According to Antibiotic Classes					
			β-Lactamases **	Amino-Glycoside	Fosfomycin	Quino-lone	Folate Pathway	Tetracycline
*Enterobacter* species								
RR26-1	*Ent. horm.* ssp. *steigerwaltii*	ST987	ACT-15	-	*fosA*/*fosA2* family	-	-	-
RR30-2	*Ent. horm.* ssp. *steigerwaltii*	**ST2689**	ACT-17	-	*fosA*/*fosA2* family	-	-	-
HC37-2	*Ent. horm.* ssp. *steigerwaltii*	ST987	ACT-15	-	*fosA*/*fosA2* family	-	-	-
MG16r-1	*Ent. horm.* ssp. *steigerwaltii*	ST2412	ACT-17	-	*fosA*/*fosA2* family	-	-	-
MG23r-3	*Ent. horm.* ssp. *steigerwaltii*	**ST2693**	ACT-17	-	*fosA*/*fosA2* family	-	-	-
MG60-4	*Ent. horm.* ssp. *steigerwaltii*	ST1439	ACT-56	-	*fosA*/*fosA2* family	-	-	-
MG65-1	*Ent. horm.* ssp. *steigerwaltii*	ST987	ACT-15	-	*fosA*/*fosA2* family	-	-	-
MG76-3	*Ent. horm.* ssp. *steigerwaltii*	**ST2696**	ACT-56	-	*fosA*/*fosA2* family	-	-	-
MG77-5	*Ent. horm.* ssp. *steigerwaltii*	**ST2697**	**ACT-122**	-	*fosA*/*fosA2* family	-	-	-
MG89-2	*Ent. horm.* ssp. *steigerwaltii*	ST1461	**ACT-123**	-	*fosA*/*fosA2* family	-	-	-
MG91-5	*Ent. horm.* ssp. *steigerwaltii*	**ST2699**	**ACT-124**	-	-	-	-	-
HC36-2	*Ent. horm.* ssp. *xiangfangensis*	**ST2691**	ACT-115	-	*fosA*/*fosA2* family	-	-	-
TM110-1	*Ent. horm.* ssp. *xiangfangensis*	**ST2692**	ACT-16	-	*fosA*/*fosA2* family	-	-	-
MG66-1	*Ent. horm.* ssp. *xiangfangensis*	**ST2692**	ACT-16	-	*fosA*/*fosA2* family	-	-	-
MG70-1	*Ent. horm.* ssp. *xiangfangensis*	**ST2692**	ACT-16	-	*fosA*/*fosA2* family	-	-	-
MG71-1	*Ent. horm.* ssp. *xiangfangensis*	**ST2692**	ACT-16	-	*fosA*/*fosA2* family	-	-	-
MG74-2	*Ent. horm.* ssp. *xiangfangensis*	**ST2692**	ACT-16	-	*fosA*/*fosA2* family	-	-	-
MG76-2	*Ent. horm.* ssp. *xiangfangensis*	**ST2692**	ACT-16	-	*fosA*/*fosA2* family	-	-	-
SS01-1	*Ent. horm.* ssp. *hormaechei*	**ST2690**	**ACT-120**	-	*fosA*/*fosA2* family	-	-	-
MG59-4	*Ent. horm.* ssp. *hormaechei*	**ST2694**	**ACT-121**	-	*fosA*/*fosA2* family ***	-	-	-
MG59-6	*Ent. cloacae* ssp. *cloacae*	**ST2695**	**CMH-13**	-	*fosA*/*fosA2* family	-	-	-
MG79-1	*Ent. quasihormaechei*	**ST2698**	ACT-59-like	-	*fosA*/*fosA2* family	-	-	-
Other *Enterobacterales* species								
MG59-3	*E. coli*	ST7588	CTX-M-15, TEM-IB	*aph(6)-Id*, *aph(3″)-Ib*, *mdf(A)*	-	*qnrS1*	*dfrA14*, *sul2*	*tet(A)*
MG86-2	*K. oxytoca*	ST19	OXY-2-18-like	-	*fosA5*/*fosA10*-like	-	-	-
MG23r-2	*Morg. morganii*	n.t.	**DHA-32**	-	*fosA*/*fosA2 family*	-	-	-
MG60-1	*Morg. morganii*	n.t.	**DHA-31**	-	*fosA*/*fosA2 family*	-	-	-
SS01-3	*S. enterica* ssp. *enterica* ^#^	ST516	-	*aac(6′)-Iaa*	-	-	-	-
SS02-5	*S. enterica* ssp. *enterica* ^#^	ST3780	-	*aac(6′)-Iaa*	-	-	-	-
ET01-4	*S. enterica* ssp. *enterica* ^#^	ST414	-	*aac(6′)-Iaa*	-	-	-	-
MG23r-5	*S. enterica* ssp. *enterica* ^#^	**ST10918**	-	*aac(6′)-Iaa*	-	-	-	-
MG77-6	*S. enterica* ssp. *enterica* ^#^	**ST10919**	-	*aac(6′)-Iaa*	-	-	-	-
MG59-1	*Serratia marcescens*	n.t.	SRT	*aac(6′)-Ic*	-	-	-	*tetA*(41)
MG91-1	*Serratia marcescens*	n.t.	SRT/SST	*aac(6′)-Ic*	*fosA*/*fosA2* family	-	-	*tetA*(41)
*Acinetobacter* spp.								
MG95-1	*A. baumannii*	ST2306^Pasteur^	ADC-25-like, OXA-91	-	-	-	-	-
SS02-1	*A. radioresistens*	n.t.	OXA-813	-	-	-	-	-
MG60-2A	*A. radioresistens*	n.t.	OXA-815	-	-	-	-	-
MG60-3	*A. radioresistens*	n.t.	OXA-815	-	-	-	-	-
MG87-2	*A. radioresistens*	n.t.	OXA-816	-	-	-	-	-
MG91-2	*A. radioresistens*	n.t.	**OXA-1221**	-	-	-	-	-
MG77-2	*A. variabilis*	n.t.	-	-	-	-	-	-

* Strain name includes abbreviation of animal source, sample I.D., and isolate number. ** Novel multilocus sequence types, as well as novel AmpC and other beta-lactamase types are written in bold. *** Gene interrupted by insertion sequence (IS) element. ^#^
*Salmonella* spp. isolates revealed chromosomal mutations in *parC*: p.T57S. ET = *Echinops telfairi*; HC = *Macronycteris* (formerly *Hipposideras*) *commersoni*; MG = *Microcebus griseorufus*; RR = *Rattus rattus*; SS = *Setifer setosus*; TM = *Triaenops menamena*; *A.* = *Acinetobacter*; *E.* = *Escherichia*; *Ent.* = *Enterobacter*; *K.* = *Klebsiella*; *Morg.* = *Morganella*; *S.* = *Salmonella*; n.t. = not typed.

## Data Availability

Data are contained within the article and Appendix A.

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
