# Peer review of "ESBL-Type and AmpC-Type Beta-Lactamases in Third Generation Cephalosporin-Resistant *Enterobacterales* Isolated from Animal Feces in Madagascar"

_animals, 2024, doi:10.3390/ani14050741_

Round 1

Reviewer 1 Report

Comments and Suggestions for Authors

The research tackles a worldwide very important subject, that of antimicrobial resistance and its triggering mechanisms, novel in animals in Madagascar. The results of such a research should add to the pool of information on AMR and help tailored development of control strategies in both human and veterinary/wildlife medicine, under One Health perspective. 

Although the sampling is thoroughly described, some elements are missing such as the selection criteria underlying the sampled species, number of individuals of each species. The ethical permit of the veterinary/wildlife responsible authorities is missing. No specifications are provided on the permit for transportation of the potentially pathogenic samples from Madagascar (other than bacterial DNA) to Germany, to the site of bacteriological analysis.

The results are presented in a very "informative" and not "participatory" manner, considering the latter one including the authors' opinion on possible associations between the habitat of a certain species, potential contact with settlements/domestic animals or the characteristics of the environment (i.e., very touristic areas for purchasing samples) as a potential cause of the presence of resistance genes.

Similarly, the Discussions could be broadened in the same sense, to underline the importance of the finding, no matter the small number of the samples investigated.

A good study, which could be much better presented for the information throughput.

Author Response

We are very grateful for the helpful comments and tried to answer all points raised by reviewer 1.

Reviewer 1: Although the sampling is thoroughly described, some elements are missing such as

a) Selection criteria underlying the sampled species

Answer: As described in the methods part, „the samples were subjected to cultural growth on agar plates selecting for Enterobacterales (CHROMagar Orientation, Mast Group Ltd.) and Acinetobacter spp. (CHROMagar Acinetobacter, Mast Group Ltd.). To verify culturable bacterial background, all samples were additionally streaked on sheep blood agar and Gassner-Agar (Merck KG, Germany). The screening for Salmonella spp. was done after selective enrichment in Rappaport-Vasiliadis-Medium (Merck KG, Germany), followed by streaking on XLD- and BPLS-Agar (Sigma Aldrich, Germany).“

Selective cultivation of samples on media containg antibiotics, such as cefotaxime or ceftazidim in order to select for ESBL- or AmpC-β-lactamase producing bacteria, was not performed. This is indeed a shortcoming of the applied methodologies, but cannot be conducted at this stage of the study as the samples are no longer available.  

To adress this, we rephrased the last paragraph of the manuscript as follows:

„The study has a few limitations. The major limitations are the relatively low sample count, the more or less arbitrary sampling pattern and the non-selective culturing approach. As such, the assessment provides just preliminary information and cannot replace future systematic analyses. The selective culturing of samples on media containing cefotaxim, ceftazidime and carbapenems might have increased the number of ESBL-, AmpC-β-lactamase and carbapenemase producing bacteria. Considering the still scarcely available epidemiological information on environmental isolates from the assessed region and with respect to the detailed molecular characterization provided for the obtained Enterobacterales and Acinetobacter isolates, the assessment may nevertheless serve as a useful proof-of-principle.“

b) Number of individuals of each species sampled

Answer: The question is already answered in the Methods chapter, sub-heading “Sample collection as follows: “fecal swabbings of a total of 49 stool droppings from Echinops telfairi (Lesser hedgehog tenrec; n=1), Macronycteris commersoni (Commerson's roundleaf bat; n=6), Microcebus griseorufus (Reddish-gray mouse lemur; n=32), Pyxis arachnoides (Spider tortoise; n=1), Rattus rattus (Black rat; n=6), Setifer setosus (Greater hedgehog tenrec; n=2) and Triaenops menamena (Rufous trident bat; n=1)“.

c) The ethical permit of the veterinary/wildlife responsible authorities is missing

Answer: The information has been provided in the Material and Methods chapter 2.1., which has been rephrased as follows: “Sample collection and Ethical permit”

“During the sampling procedure, compliance with all applicable institutional and/or national guidelines for the care and use of animals was assured. Ethical clearance for the field work and permit for the sample shipment were provided by the ethics committee of the Institute of Zoology of Hamburg University before the initiation of this study and authorized by the Ministère de l'Environnement, de l'Ecologie, de la Mer et des Forêts (Research permits: N°136/16/- and N°002/ 17/MEEF/SG/ DGF/DSAP/SCB.Re; export permit: N°345-17/MEEF/SG/DGF/ DREEFAAND/SFR).”

d) No specifications are provided on the permit for transportation of the potentially pathogenic samples from Madagascar (other than bacterial DNA) to Germany, to the site of bacteriological analysis.

Answer: The information on the export permit (export permit: N°345-17/MEEF/SG/DGF/ DREEFAAND/SFR)) has been included in the new “Ethical permit” section.

e) The results are presented in a very "informative" and not "participatory" manner, considering the latter one including the authors' opinion on possible associations between the habitat of a certain species, potential contact with settlements/domestic animals or the characteristics of the environment (i.e., very touristic areas for purchasing samples) as a potential cause of the presence of resistance genes. Similarly, the Discussions could be broadened in the same sense, to underline the importance of the finding, no matter the small number of the samples investigated.

Answer: As requested, the following new text-block on this topic has been added to the discussion:

 “Altogether, the proportion of detected resistant bacteria within the assessed animal stool samples is low to moderate and can thus be well explained by occasional contacts with the Madagascan human civilization that is much more severely affected by the resistance issue. In particular, only individual cases of ESBL-type resistance have been detected in the animal stool samples of the here-presented study, while the ESBL mechanism has been described to account for high colonization rates with 3rd generation cephalosporin-resistant Enterobacterales in both Madagascan people and livestock [7,14-18,22,23]. The considerable higher rates of colonization with ampC-positive Enterobacter spp. is well in line with observations in Madagascan patients and healthcare workers, in which this genus-resistance type-combination accounted for a major part of recorded third generation cephalosporin resistance in isolated colonizing Enterobacterales [18]. Insofar, their regional abundance seems to be typical for Madagascar. Finally and with focus on the isolated Acinetobacter spp., recorded resistance profiles were close to the wild type situation and in particular, carbapenem resistance-mediating genes like previously described for the Madagascan setting were not recorded. Altogether, the observed resistance profiles matches the expectations from the literature quite well.”

Reviewer 2 Report

Comments and Suggestions for Authors

A good study on the prevalence of ESBL-type and AmpC-type beta-lactamases in third generation cephalosporin-resistant Enterobacterales isolated from animal feces in Madagascare, only minor revisions are needed.

Subsection 2.2. How many isolates were chosen for confirmation from each sample/agar?

Line 159. From how many samples were 41 bacterial isolates obtained?
I recommend showing the prevalence of bacterial isolates (3.1.) in a table.

This is an important information but difficult to follow now.

Line 372. Fosfomycin.

Author Response

Thank you very much for your comments. We hope that our answers are addressing them properly. 

Reviewer 2: Subsection 2.2. How many isolates were chosen for confirmation from each sample/agar?

Answer: As requested, we have now stated that for each different colony morphology as visible with the bare eye, a single colony was chosen for further assessments (Section 2.2).

Reviewer 2: From how many samples were 41 bacterial isolates obtained? I recommend showing the prevalence of bacterial isolates (3.1.) in a table. This is an important information but difficult to follow now.

Answer: The recommended new table (see Table 1 in the revised manuscript) has been added as requested. The former tables have been renumbered as required.

Reviewer 2: Line 372. Fosfomycin.

Answer: The typing error has been corrected

Reviewer 3 Report

Comments and Suggestions for Authors

The study aimed to investigate the fecal carriage of extended-spectrum cephalosporin-resistant bacteria in wild animals in Madagascar, utilizing both conventional and next-generation sequencing tools.

Despite the critical need for antimicrobial resistance (AMR) data in Africa, including Madagascar, the primary drawback of this investigation lies in the inadequacy of the chosen methodology. To enhance the reliability of their findings, the authors should have employed a more selective media supplemented with ceftazidime or cefotaxime.

Under the current methodology, the authors fall short of screening for resistance, limiting their analysis to characterizing growth colonies on the adopted culture media. While the paper holds significance for public health, the methodology's inherent shortcomings undermine its potential contribution to drawing robust conclusions.

A noteworthy example is the identification of only one animal harboring a CTX-M-producing isolate, which may not accurately depict the prevailing situation. The adoption of a more suitable methodology could potentially reveal a higher number of instances.

Furthermore, the detection of fosA presence is a cause for serious concern and warrants attention.

In conclusion, while the paper addresses an important public health concern, refining the methodology is crucial for the study to fulfill its potential in contributing valuable insights.

Comments on the Quality of English Language

NA

Author Response

We are very grateful for the comments and suggestions given by reviewer 3. We hope, that the performed changes address the points raised sufficently.

Reviewer 3: …….the primary drawback of this investigation lies in the inadequacy of the chosen methodology. To enhance the reliability of their findings, the authors should have employed a more selective media supplemented with ceftazidime or cefotaxime.

Answer: We agree with the reviewer that a more selective approach could have revealed a higher number of AMR bacteria. Unfortunately, the samples are no longer available and additional investigations as proposed by the reviewer cannot be performed. Nevertheless, we think that the results in its current form still hold significance for public health. However, as we cannot validly estimate how the results from our non-selective approach would differ from the suggested selective approach, we rephrased the last paragraph of the manuscript as follows:

„The study has a few limitations. The major limitations are the relatively low sample count, the more or less arbitrary sampling pattern and the non-selective culturing approach. As such, the assessment provides just preliminary information and cannot replace future systematic analyses. The selective culturing of samples on media containing cefotaxime and ceftazidime might have increased the number of isolated AMR bacteria. Thus, the data cannot provide a real estimate for the distribution of ESBL- and AmpC-β-producing bacteria among samples from the given animal population. Considering the still scarcely available epidemiological information on environmental isolates from the assessed region and with respect to the detailed molecular characterization provided for the obtained Enterobacterales and Acinetobacter isolates, the assessment may nevertheless serve as a useful proof-of-principle.

Reviewer 3: Furthermore, the detection of fosA presence is a cause for serious concern and warrants attention.

Answer: Thanks for this good point. We re-checked our in silico screening for the presence and type of fosA genes in all genomes (also by manual search using Genious). This confirmed the presence of fosA in the Enterobacter spp. isolates as indicated in the intitial version of the manuscript and revealed its additional presence in two Morganella morganii isolates, in Serratia marcescens isolate MG91-1 and in the K. oxytoca isolate (see amended Table 4 [former Table 3].   

An evolutionary analysis and phylogenetic tree of all publicly available FosA/C2/L1-2 and protein sequences and of the FosA sequences identified in our study was inferred by using the Maximum Likelihood method using Geneious Prime® 2023.2.1 to determine the FosA type, which was FosA/FosA 2 (genes chromosomally located) family in nearly all cases.

We have added a few sentences in section 3.2 and 4. as well as a novel Figure (see Figure 2 in the revised manuscript version) in section 3.2.

Section 3.2:

„Nearly all Enterobacter spp. isolates as well as the two M. morganii isolates, S. marcescens isolate MG91-1 and K. oxytoca isolate MG86-2 carried fosfomycin resistance gene fosA. An evolutionary analysis (Figure 2) revealed the assignment of Enterobacter spp. FosA proteins to the FosA2 family, which is correlated with a chromosomal location of the fosA gene. FosA proteins of M. morganella and S. marcescens isolates were genetically highly related or identical to FosA reference proteins of the same species, not clearly labelled with an allele number. The FosA protein of K. oxytoca isolate MG86-2 revealed the highest similarity to FosA5 (88.5%) and FosA10 (88.5%). Also for the non-Enterobacter sp. isolates, there was no indication for a plasmid location of fosA.

Section 4:

However, the high percentage of isolates carrying a fosA gene, which confers resistance to fosfomycin, was not to be expected. This old antibiotic regained relevance in clinical practice for the treatment of complicated infections caused by multi-drug resistant bacteria [31]. Therefore, the emergence of fosA among rather naïve wild animal populations warrants further investigations as it might have a significant impact on public health.

Round 2

Reviewer 1 Report

Comments and Suggestions for Authors

The authors improved their first version of the manuscript, one question was still unanswered - that on how the animals species to be investigated (and not the bacteria) were selected? Presumably, the authors samples those species which they has at hand or those which were the most frequently present.

Author Response

Thanks for reminding us to comment on this. We inserted the following text into the methods section and hope that this sufficiently answers your question:

"The animals caught at the study site were subjected to sampling without particular selection criteria. Insofar, the species distribution indirectly reflects their local abundance at the study site or at least their affinity to the applied live traps."

Reviewer 3 Report

Comments and Suggestions for Authors

The authors significantly improved their manuscript, particularly after adding the limitation section and mentioning the non-use of selective media. This latter is required to screen for MDR pathogens.

Comments on the Quality of English Language

NA

Author Response

Thanks!